# In Vitro Hepatotoxicity of Routinely Used Opioids and Sedative Drugs

Katharina Haller [1,†], Sandra Doß [2,†] and Martin Sauer [2,3,4,*]

[1] Department of Anesthesiology and Intensive Care Medicine, Charité Campus Benjamin Franklin, Hindenburgdamm 30, 12203 Berlin, Germany; katharina.haller@charite.de
[2] Department Extracorporeal Therapy Systems (EXTHER), Fraunhofer Institute for Cell Therapy and Immunology, Schillingallee 68, 18057 Rostock, Germany; sandra.doss@izi.fraunhofer.de
[3] Department of Anesthesiology and Intensive Care Medicine, University Hospital of Rostock, Schillingallee 35, 18057 Rostock, Germany
[4] Center for Anesthesiology and Intensive Care Medicine, Hospital of Magdeburg, Birkenallee 34, 39130 Magdeburg, Germany
* Correspondence: martin.sauer@uni-rostock.de
[†] These authors contributed equally to this work.

**Abstract:** A hepatocyte cell line was used to determine the hepatotoxicity of sedatives and opioids, as the hepatotoxicity of these drugs has not yet been well characterized. This might pose a threat, especially to critically ill patients, as they often receive high cumulative doses for daily analgosedation and often already have impaired liver function due to an underlying disease or complications during treatment. A well-established biosensor based on HepG2/C3A cells was used for the determination of the hepatotoxicity of commonly used sedatives and opioids in the intensive care setting (midazolam, propofol, s-ketamin, thiopental, fentanyl, remifentanil, and sufentanil). The incubation time was $2 \times 3$ days with clinically relevant (Cmax) and higher concentrations (C5× and C10×) of each drug in cell culture medium or human plasma. Afterward, we measured the cell count, vitality, lactate dehydrogenase (LDH), mitochondrial dehydrogenase activity, cytochrome P 450 1A2 (CYP1A2), and albumin synthesis. All tested substances reduced the viability of hepatocyte cells, but sufentanil and remifentanil showed more pronounced effects. The cell count was diminished by sufentanil in both the medium and plasma and by remifentanil only in plasma. Sufentanil and remifentanil also led to higher values of LDH in the cell culture supernatant. A reduction of mitochondrial dehydrogenase activity was seen with the use of midazolam and s-ketamine. Microalbumin synthesis was reduced in plasma after its incubation with higher concentrations of sufentanil and remifentanil. Remifentanil and s-ketamine reduced CYP1A2 activity, while propofol and thiopental increased it. Our findings suggest that none of the tested sedatives and opioids have pronounced hepatotoxicity. Sufentanil, remifentanil, and s-ketamine showed moderate hepatotoxic effects in vitro. These drugs should be given with caution to patients vulnerable to hepatotoxic drugs, e.g., patients with pre-existing liver disease or liver impairment as part of their underlying disease (e.g., hypoxic hepatitis or cholestatic liver dysfunction in sepsis). Further studies are indicated for this topic, which may use more complex cell culture models and global pharmacovigilance reports, addressing the limitation of the used cell model: HepG2/C3A cells have a lower metabolic capacity due to their low levels of CYP enzymes compared to primary hepatocytes. However, while the test model is suitable for parental substances, it is not for toxicity testing of metabolites.

**Keywords:** analgosedation; drug-induced liver injury (DILI); hepatotoxicity; intensive care; opioids; sedatives

## 1. Introduction

Providing adequate sedation and analgesia for critically ill patients is essential [1]. Therefore, many patients receive sedatives and analgesics (often opioids) during their stay in an intensive care unit (ICU). There is also increasing evidence that the overuse of these drugs leads to an increase in mortality and prolonged ICU and hospital stays [2–4]. Opioids are drugs used to treat severe pain. Examples of opioids are buprenorphine, codeine, oxycodone, tapentadol, and tramadol, which are used for the treatment of severe pain [5]; sufentanil, fentanyl, morphine, and remifentanil are used for sedation and analgesia in ICU patients too [6]. The adverse effects of opioids are known to include delirium, bowel dysfunction, and ICU-acquired infections [7]. ICU patient sedation is achieved by drugs from different groups like benzodiazepines (lorazepam and midazolam) and the short-acting intravenous anesthetics propofol, s-ketamine, and barbiturate [5,8]. The adverse effects of these drugs differ widely, ranging from delirium, possible organ toxicity, and hypotension to a disruption of the mitochondrial respiratory chain (propofol infusion syndrome) [5,8]. Current sedation management has become more complex. With the development of new drugs, e.g., alpha-2-agonists, in combination with lower concentrations of "classical" sedation drugs are used [6].

Aside from these known adverse effects, there is also a higher risk of drug toxicity for critically ill patients due to their hemodynamic instability or impaired organ function, which leads to altered pharmacokinetics and pharmacodynamics [9,10]. Drug-induced liver injury is a significant cause of acute liver failure, and the mortality rate in patients with acute liver failure is approximately 80% [11]. Concerning drug-induced liver injury (DILI), little is known about its impact on critically ill patients. The reason for this could be that in ICUs, many factors can result in liver injury: an underlying disease, a complication during treatment (e.g., sepsis), the administration of drugs over a long period and in sometimes high doses, or a combination of multiple factors. Opioid-based medications and sedatives may interact in combination with hepatotoxic medications (e.g., acetaminophen). This increases the risk of liver damage if they are administered at the same time [12].

Opioids are relatively rare causes of drug-induced liver disease. Still, overdoses of the more potent opioids have been associated with cases of acute liver injury, which have a sudden onset and a pattern of acute toxicity, with marked elevations in serum aminotransferase levels and an early onset of signs of liver failure. After cannabis, opioids and cocaine are the most commonly consumed drugs [12–15].

The liver is the primary target of drug-induced toxicity, and hepatotoxicity is an important endpoint in the safety assessment of drugs and chemicals. Therefore, assessing potential hepatotoxicity represents a crucial step in developing new drugs [16]. In general, DILI is a difficult field of research because most cases are unpredictable, idiosyncratic, and rare, making them difficult to study. As a result, there has been limited progress in controlling, understanding, or preventing DILI over the past 50 years. Several methods have been developed to improve the assessment of the causes of hepatotoxicity because diagnosis remains a major challenge. These species-specific causality assessment tools fall into three categories: (1) probabilistic approaches, (2) expert judgments, and (3) algorithms or scales [17,18]. However, there is generally an urgent need for models that predict hepatotoxicity in humans [19].

Clinical trials are generally underpowered to detect trends in hepatotoxicity, so case reports of adverse drug reactions are the primary source of toxicity data [20]. Current strategies for testing the potential for DILI rely on in vivo animal models. While animal-based toxicity testing was able to predict 70% of the experienced toxicity in humans in a retrospective analysis, human DILI is difficult to predict in animals, likely due to known interspecies differences in drug metabolism, pharmacokinetics, and toxicity targets [19,21–24]. Therefore, the collection of phenotypic information from appropriate registries and biological samples from identified DILI cases is currently the most valuable resource for reproducing the complexity of idiosyncratic DILIs [25]. LiverTox (https://livertox.nih.gov, accessed on

10 January 2024), for example, is a clinical research database that provides an overview of possible drug-induced hepatotoxicity. This database was developed by the Drug-Induced Liver Injury Network (DILIN) and is published in NCBI's LiverTox. The LiverTox Likelihood Score is used to categorize the likelihood that a drug is associated with drug-induced liver injury. It also includes a description of the pattern and history of liver injury, as well as case studies supported by laboratory data and a comprehensive list of references [26].

There is an urgent need for human-relevant in vitro models for preclinical testing during drug development processes [27]. In recent years, several in vitro systems have been developed for toxicological applications [21]. We used an established and standardized biosensor [28] with the well-established immortalized human HepG2/C3A cells, a patented, highly functional clonal derivative of the human hepatoma cell line HepG2, to test the hepatotoxicity of routinely used sedatives and opioids [29]. Evaluating multiple endpoints on Hep G2 cells allows the prediction of human hepatotoxicity with a sensitivity of over 80% and a specificity of 90% [30]. Accordingly, HepG2 cells and derivatives can predict general human hepatotoxicity based on hepato-specific endpoints [19,31]. These cell lines are characterized by their functional similarity to the human liver, physiological response to toxic insults, and metabolic markers [28,29,32–40]. HepG2/C3A is able to synthesize most plasma proteins, including albumin [41]. However, their main disadvantage is their lower metabolic capacity due to low levels of CYP enzymes compared to primary isolated hepatocytes. This makes them suitable test models for the parental substances but not for the toxicity of metabolites [42]. HepG2/C3A is able to synthesize most plasma proteins, including albumin [41]. They produce bile acids as well as glycogen and express many hepatic functions, such as cholesterol and triglyceride metabolism, lipoprotein metabolism, or insulin signaling [42–44]. Differences are, however, a non-functional urea cycle, low levels of phase-II enzymes (sulfotransferase, uridine diphosphate glucuronosyltransferase, glutathione S-transferase, or N-acetyltransferase), and transport proteins (organic anion transporting polypeptide C, bile salt export pump, and sodium-taurocholate co-transporting polypeptide) [45]. Although there are low or absent basal levels of important CYP enzymes (such as CYP3A4, CYP2C9, CYP2C19, CYP2A6, or CYP2D6) compared to primary hepatocytes [46], similar inducibility has been shown for CYP 1A1, 1A2, 2B6, and 3A4 [44,47]. On the other hand, primary isolated human hepatocytes are scarcely available, exhibit inconsistent characteristics, and show CYP induction that varies from donor to donor, which may depend on the patient's medication.

The aim of the study was the testing of hepatotoxicity using clinically relevant concentrations of drugs during continuous analgosedation (steady-state plasma concentration).

## 2. Materials and Methods

### 2.1. Cell Culture

Human hepatocellular carcinoma cells (HepG2/C3A, ATCC, ref. number CRL-10741) maintained in Dulbecco's modified Eagle's medium (DMEM, GIBCO Life Technologies, Darmstadt, Germany), supplemented with 10% fetal bovine serum (FBS, Biochrome, Berlin, Germany), 1% of 200 mM L-glutamine (Biochrome), and 1% of antibiotics solution (Penicillin G:10.000 IE/mL/Streptomycin: 10 mg/mL; Biochrome, Berlin, Germany) were used. Cells were routinely incubated under a humidified atmosphere containing 5% $CO_2$ at 37 °C and were regularly subcultured every 2–3 days.

### 2.2. Drug Solutions

A Cmax-based testing approach is a useful strategy to distinguish between safe and hepatotoxic drugs; moreover, a multiple of the maximum plasma concentration (C max) of the drugs studied was used because some hepatotoxic drugs show significant cytotoxicity in the 10 to 100-fold Cmax range [21,43]. Therefore, the lowest test concentration of the various sedative medications was the mean plasma level after initiation of continuous intravenous therapy (C max, steady-state plasma concentration) based on clinical application data;

additionally, to Cmax, we tested two higher concentrations of the drugs (5× Cmax, 10× Cmax): midazolam (300, 1500, and 3000 ng/mL; mmol; ratiopharm, Ulm, Deutschland) [44], propofol (2, 10, and 20 µg/mL; mmol; B.Braun, Melsungen, Deutschland) [45,46], s-ketamin (1, 5, and 10 µg/mL; mmol; Pfizer, New York City, NY, USA) [47], thiopental (1, 5, and 10 mg/mL; mmol; Sandoz, Holzkirchen, Germany) [48], fentanyl (10, 50, and 100 ng/mL; Janssen-Cilag GmbH, Neuss, Deutschland) [49], remifentanil (15, 75, and 150 ng/mL; mmol; GSK, München, Deutschland) [50,51], and sufentanil (1, 5, and 10 ng/mL; mmol; Janssen-Cilag GmbH, Neuss, Deutschland). All drugs were dissolved and diluted with water, whereas propofol had to be dissolved with Dimethyl sulfoxide (DMSO) (0.1% (*v/v*)). It was important to optimize the DMSO concentration in the cell culture experiments and to consider the specific requirements of the cells used in order to evaluate the possible effects or influences of DMSO itself. Appropriate control groups were included in the experiments to compare the effects of DMSO-treated samples at a concentration of 0.9% with untreated ones, and no negative effects on the cells could be observed (see Table S1, Supplement).

*2.3. Treatment and Cytotoxicity Assay*

The cell treatment procedure was conducted according to an established microtiter plate assay for the screening of hepatotoxicity [28,33–40]: HepG2/C3A cells ($5 \times 10^5$) were plated in a 24-well plate and the cells were exposed to different concentrations (Cmax, 5× Cmax, 10× Cmax) of the several sedating drugs (see Section 2.2) for 72 h in the medium or heparinized plasma from healthy volunteers (pooled plasma). Afterward, the cells were treated with the three concentrations of the drugs only in the culture medium for 72 h again. The negative controls served cells, which were incubated with the medium or plasma (without agents). Each assay was performed in triplicate, and the experiments were repeated at least five times.

The pH values (Radiometer, ABL, Willich, Germany) were screened at the beginning of each experiment in the cell culture supernatant, and a normal range was found in the medium and plasma (see Tables S2 and S3, Supplement).

The tests commonly used to evaluate cytotoxicity are the colorimetric assay with 2,3-bis(2-methoxy-4-nitro-5-sulfophenyl)-S-(phenylamino) carbonyl-2-tetrazolium hydroxide (XTT) and the trypan blue exclusion assay [52,53]. The XTT assay, which determines the metabolic activity and proliferation of cells, was performed according to the manufacturer's instructions (XTT, Roche Diagnostics GmbH, Mannheim, Germany) [54]. XTT is a tetrazolium salt that cleaves to formazan by the succinate dehydrogenase system, which belongs to the mitochondrial respiratory chain. This is significant, as it is only active in viable cells. The optical density (OD) was measured on a microplate reader (Anthos Reader 2001, Anthos Labtec Instruments, Austria) at 450 nm after one hour of assay time. The trypan blue exclusion test (0.4% (*w/v*); Sigma, Seelze, Germany) rapidly assesses cell viability. The test is based on the principle that viable cells with intact cell membranes are not colored, while trypan blue traverses the membrane in a dead cell [55]. The total cell counts/mL were determined in a Neubauer-improved cell-counting chamber (peqlab, Erlangen, Germany) by manually counting the number of colored (dead) cells and unstained (viable) cells with light microscopy.

Albumin production of the cultured hepatocytes was examined nephelometrically from 0.2 mL of the cell culture medium supernatant (Immage 800, Beckman Coulter GmbH, Krefeld Germany) [56].

Lactate dehydrogenase (LDH) levels in the supernatants were photometrically analyzed by the change in the absorbance at 340 nm with the automated chemistry analyzer (Cobas Mira, Roche, Mannheim, Germany) according to the optimized standard method of the Deutsche Gesellschaft für Klinische Chemie (DGKC) after 144 h of incubation [57].

The enzyme 7-ethoxy-resorufin-O-deethylase (EROD) activity was performed as described by Donato et al. [58]. Before the measurement, hepatocytes were stimulated with methylcholanthrene (3-MC, Sigma Aldrich, Seelze, Germany) for 72 h. The EROD assay

was initiated by incubating the stimulated hepatocytes with 8 μM 7-ethoxyresorufin (Molecular Probes, Eugene, OR, USA) and 10 μM dicumarol (Sigma Aldrich, Seelze, Germany) for 1 h (37 °C, 5% $CO_2$) in the culture medium. After 1 h of incubation under lightproof conditions, the fluorescence of the metabolites from 7-ethoxy-resorufin by HepG2/C3A cells was detected at 530 nm (excitation) and 584 nm (emission) by using a fluorescence multiwell plate reader (Fluoroskan Ascent Lab Systems, Vienna, VA, USA). A resorufin standard curve (0–80 pmol) was included for each plate.

### 2.4. Statistics

Statistical analysis was performed using SPSS Software Version 20 (SPSS Inc., Chicago, IL, USA). Analysis among groups was performed using the Kruskal–Wallis test because data were not normally distributed. Finally, the Mann–Whitney U test for pairwise comparison was used. The data are presented as the median, 25th, and 75th quartiles with SPSS Software Version 20 (SPSS Inc., Chicago, IL, USA). Significance was accepted at a $p < 0.05$ and is indicated with an "×" in the figures in Section 3.

## 3. Results

### 3.1. Sufentanil Reduces Cell Count

Only sufentanil significantly reduced the cell count at the clinically relevant concentration (Cmax) in the cell culture medium and plasma (the results for the Cmax in the medium are shown in Figure 1A). After incubation with sufentanil, the cell count was reduced to $485 \times 10^3$/mL (medium) and $328 \times 10^3$/mL (plasma) compared to the negative control (NC) ($683 \times 10^3$/mL and $650 \times 10^3$/mL, respectively). Remifentanil reduced the cell count in plasma ($483 \times 10^3$/mL). No concentration-dependent effects were observed.

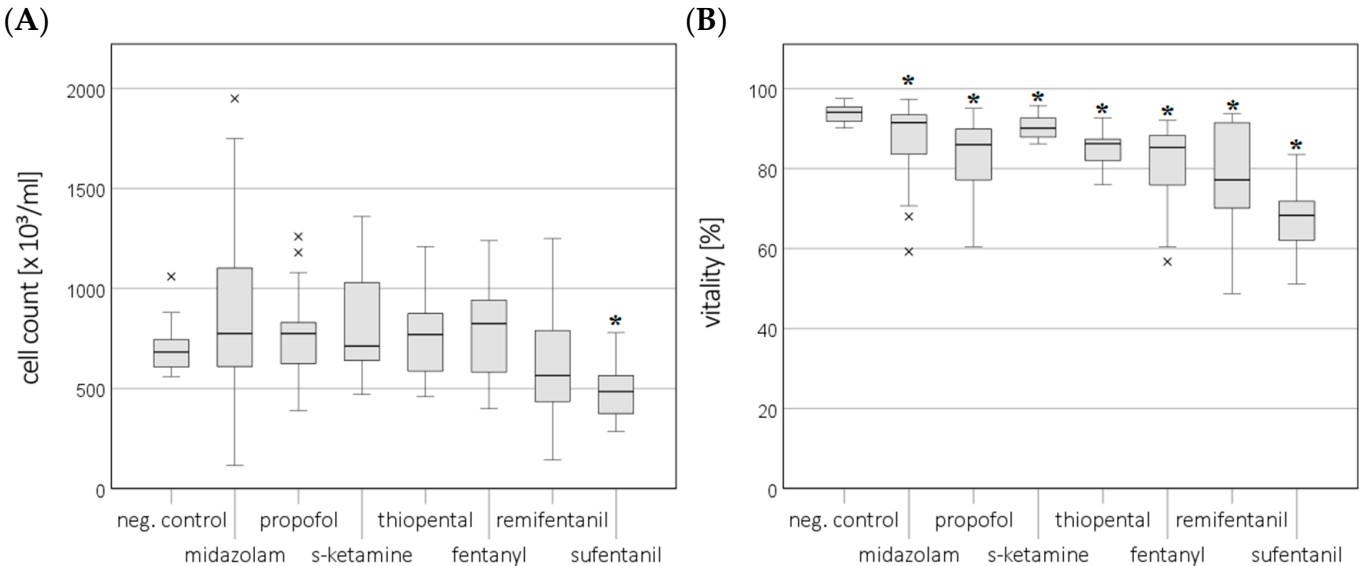

**Figure 1.** *Cont.*

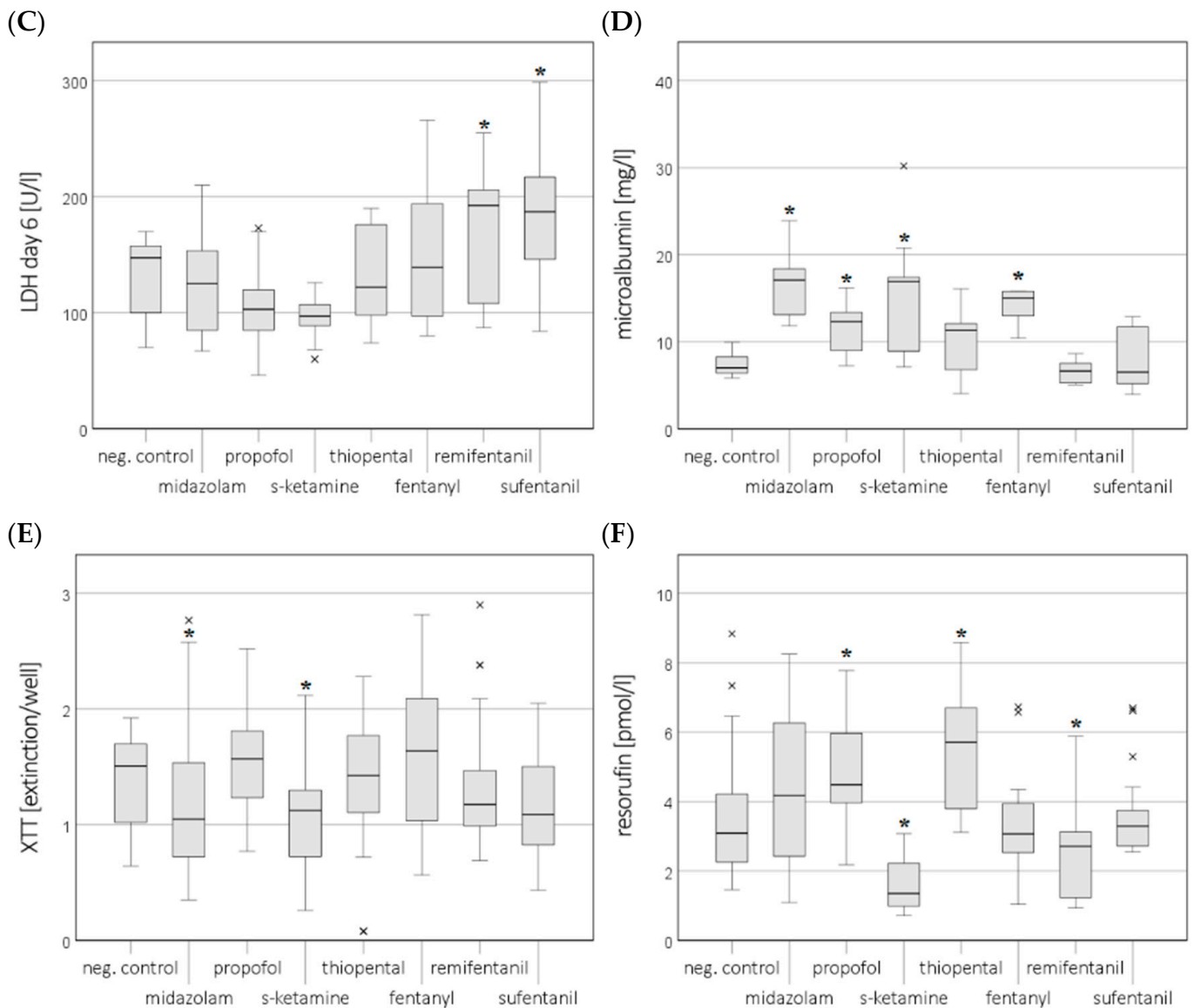

**Figure 1.** Test results for HepG2/C3A cells after exposure to Cmax (clinically relevant concentration) of sedatives and opioids in a medium. Cell count (**A**), vitality (trypan blue staining) (**B**), release of lactate dehydrogenase (LDH) (**C**), microalbumin concentration (**D**), activity of mitochondrial dehydrogenases (XTT test) (**E**), and CYP1A2 activity (resorufin concentration) (**F**). Values represent the median and 25th/75th percentiles. The significance between negative control (neg. control) and exposure groups is indicated by * $p < 0.05$, and × indicates outliers.

### 3.2. Sufentanil Causes Highest Reduction in Vitality

Vitality in the cell culture medium or plasma was significantly decreased at the Cmax for sufentanil (68%/69%), remifentanil (77%/80%), fentanyl (85%/82%), thiopental (86%/81%), and propofol (86%/77%) compared to the NC (94%/90%) (the results for the Cmax in the medium are shown in Figure 1B). A concentration-dependent effect was only recorded for thiopental in plasma, with a decrease from 81% (Cmax) to 70% (C5×) and 67% (C10×) (Figure 2).

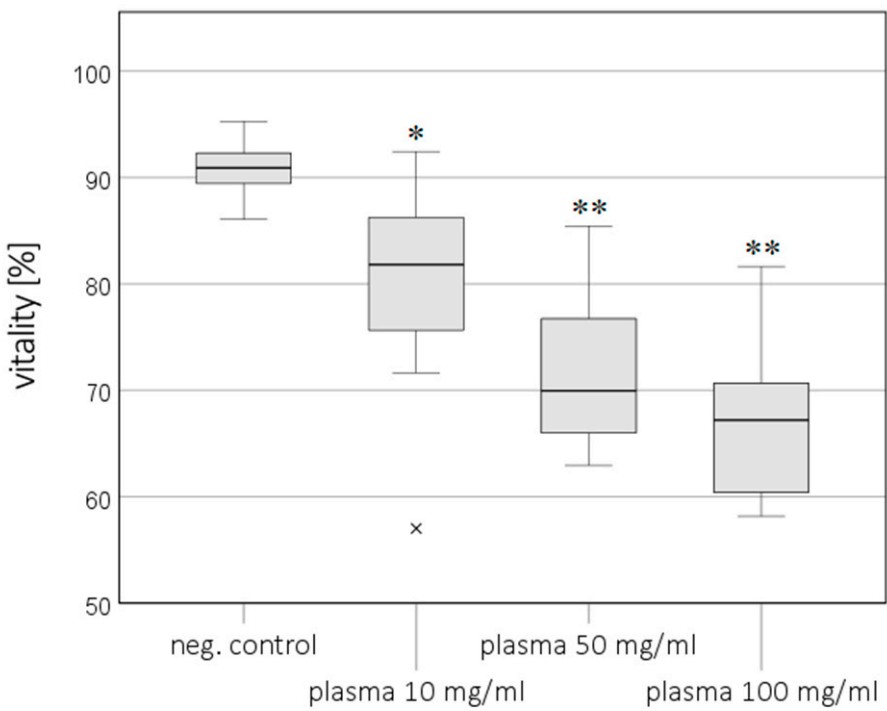

**Figure 2.** Vitality (trypan blue staining) of HepG2/C3A cells after exposure to thiopental in concentrations of 10 mg/mL (Cmax), 50 mg/mL (5× Cmax), and 100 mg/mL (10× Cmax) in plasma. Values represent the median and 25th/75th percentiles. Significance between negative control (neg. control) and exposure groups is indicated by * $p < 0.05$; significance between neg. control as well as Cmax and exposure groups are indicated by ** $p < 0.05$, and × indicates outliers.

### 3.3. Remifentanil and Sufentanil Cause Higher Release of Lactate Dehydrogenase

In the medium, after 6 days, the lactate dehydrogenase (LDH) levels were significantly increased at the Cmax after incubation with sufentanil (187 U/L) and remifentanil (193 U/L) compared to the NC (148 U/L) (the results for the Cmax in the medium are shown in Figure 1C) in the cell culture supernatant. There was no concentration-dependent effect. In plasma, there was no significantly increased LDH release after 6 days for any of the drugs.

### 3.4. Sufentanil and Remifentanil Reduce Microalbumin Synthesis

A reduced microalbumin concentration in the cell culture supernatant compared to the NC was not present in the medium at the Cmax concentrations for any of the compounds (the results for the Cmax in the medium are shown in Figure 1D). After incubation with higher concentrations of sufentanil and remifentanil (C5× − C10×), the albumin concentration in the medium decreased (4.5 mg/L and 4.7 mg/L, respectively) compared to the negative control (NC; 7.0 mg/L). In plasma, a significant reduction after incubation with all three concentrations (Cmax, C5×, and C10×) of sufentanil (6.2 mg/L, 7.2 mg/L, and 8.1 mg/L) was seen compared to the NC (13.7 mg/L).

### 3.5. Midazolam and S-Ketamine Reduce Activity of Mitochondrial Dehydrogenases

The extinction (corresponding to the activity of mitochondrial dehydrogenases in the XTT assay) was 1.5 for the NC in a medium. Midazolam (1.1) and s-ketamine (1.1) showed a significantly reduced activity of mitochondrial dehydrogenases at the Cmax concentrations (the results for the Cmax in the medium are shown in Figure 1E). There was no concentration-dependent effect. No significant reduction in the XTT assay was observed in plasma for any of the drugs.

### 3.6. S-Ketamine and Remifentanil Reduce CYP1A2 Activity

A significant reduction in resorufin concentration (CYP1A2 enzyme activity) at the $C_{max}$ compared to the NC (3.1 pmol/L) was observed after incubation with s-ketamine (1.4 pmol/L) and remifentanil (2.7 pmol/L) (the results for the $C_{max}$ in the medium are shown in Figure 1F). These effects were not concentration-dependent. A significant increase in CYP1A2 activity was seen after incubation with propofol (4.5 pmol/L) and thiopental (5.7 pmol/L). In plasma, CYP1A2 enzyme activity was significantly decreased compared to the NC (3.0 pmol/L) for midazolam (1.7 pmol/L), propofol (1.7 pmol/L), s-ketamin (1.6 pmol/L), fentanyl (1.0 pmol/L), remifentanil (1.6 pmol/L), and sufentanil (2.2 pmol/L) at the $C_{max}$ concentrations. No concentration-dependent effects in plasma were seen.

In conclusion of the results, Table 1 summarizes the parameters for estimating the hepatotoxic potential. The number of stars represents the degree of hepatotoxicity in terms of impairment parameters and in relation to the negative control (NC) without a specific ranking of the parameters among themselves.

**Table 1.** Comparison of tested sedatives and opioids in medium (M) and in plasma (P) at Cmax (clinically relevant concentration) compared to NC (neg.Ctrl.).

| | Neg.Ctrl. | | Midazolam | | Propofol | | S-Ketamine | | Thiopental | | Sufentanil | | Remifentanil | | Fentanyl | |
|---|---|---|---|---|---|---|---|---|---|---|---|---|---|---|---|---|
| | M | P | M | P | M | P | M | P | M | P | M | P | M | P | M | P |
| Cell Count (×1000) | 683 | 650 | | | | | | | | | ↓ | ↓ | ↓ | | | |
| Vitality [%] | 94 | 90 | ↓ | | ↓ | ↓ | ↓ | ↓ | ↓ | ↓ | ↓ | ↓ | ↓ | ↓ | ↓ | ↓ |
| LDH [U/L] | 148 | 168 | | | | | | | | | ↑ | | ↑ | | | |
| XTT (OD) | 1.5 | 0.8 | ↓ | | | | ↓ | | | | | | | | | |
| MA [mg/L] | 7 | 14 | | | | | | | | | | | | ↓ | | |
| CYP1A2 [pmol/L] | 3 | 3 | | ↓ | | ↓ | ↓ | ↓ | | | | ↓ | ↓ | ↓ | | ↓ |
| Level of hepatotoxicity | | | •• | • | • | •• | ••• | •• | • | • | ••• | ••• | •••• | ••• | • | •• |

The arrows show a significant ($p < 0.05$) increase (↑) or decrease (↓) in the corresponding category compared to the negative control. The number of dots corresponds to the number of arrows of all assays and represents the hepatotoxic potential of the different drugs in the medium and plasma.

## 4. Discussion

The diversity of concentration strategies makes it challenging to establish consensual concentration criteria for classifying drugs as hepatotoxic ones. Typically, steady-state drug concentrations in plasma (C ss) or maximum plasma drug concentrations (C max) are commonly used; however, they are valid strategies to differentiate between hepatotoxic and safe drugs. These test approaches do not consider the possible accumulation of drugs in the liver or protein binding. Therefore, in this study, we used multiples of the maximum plasma concentration (Cmax) (5×, 10×) of the investigated drugs to assess the toxic potential [59]. Xu et al. [43] reported that the 10 to 100-fold C max scaling factor represented a reasonable threshold to differentiate safe versus hepatotoxic drugs. Nevertheless, C max values can be easily measured and are accessible for reference compounds [21,60].

Acute liver failure (ALF) is often drug-related [61]. In preclinical models (in vitro or in rodents) and clinical trials, drug-induced hepatotoxicity (DILI) is rarely recognized [62]. Patients in intensive care might be particularly vulnerable to hepatotoxic drugs as they often already have liver damage as part of the underlying disease (e.g., hypoxic hepatitis or liver dysfunction in sepsis). As drugs for sedation and analgesia are given over a long period of time, the question of possible hepatotoxicity arises here in particular. Additionally, all

tested sedatives and opioids are hepatically metabolized mostly via CYP, with the exception of remifentanil, which can lead to the accumulation of these drugs in the course of liver insufficiency. On the other side, ICU patients often need 8 to 12 different drugs, and a possible toxic interaction and summarization of the used compounds may lead to a higher risk of organ toxicities.

In this study, commonly used sedatives and opioids (midazolam, propofol, thiopental, s-ketamine, fentanyl, remifentanil, and sufentanil) were tested for hepatotoxicity. An established cytotoxicity screening model based on HepG2/C3A cells was used to investigate the effects on cell number and vitality, LDH release, mitochondrial function (XTT assay), microalbumin synthesis, and activity of CYP1A2 [28,33–37,39,40]. However, none of the tested drugs showed pronounced cytotoxicity (a reduction of cell count by at least 50%) [63], and all tested substances reduced hepatocyte vitality (summary, see Table 1). Three drugs (sufentanil, remifentanil, and s-ketamin) presented a significant reduction in three or more of the six test parameters that could be interpreted as a moderate hepatotoxic effect (Table 1).

Animal testing is used in pharmaceutical and industrial research to predict human toxicity. This is especially true since the modulation of the immune system and cross-communication between organs cannot be achieved by developing DILI in vitro researchers based on cells and tissues [64]. Nevertheless, the animal models poorly predict human drug safety [65]. More and more researchers are questioning the scientific value of costly animal experiments [66]. Most studies evaluating DILI use rats or mice as animal models to determine drug toxicity; however, a drug's toxicity may differ in rats or mice. These inconsistencies in the in vivo studies between different animal models affect the extrapolation of experimental results to humans [67]. As alternatives to animal testing, several in vitro and in silico methods or organ-on-chip technologies have been developed that use human liver cells or tissue sections as potential screening assays to identify hepatotoxic substances [68]. They are cost-effective and provide the rapid performance of a robust safety assessment for many chemicals with limited toxicological information [69–71]. Unfortunately, the necessary standardization is lacking in DILI in vitro studies. There is no clear consensus on which models can most accurately predict hepatotoxicity in humans. All models have strengths and weaknesses, but no particular model is suitable for detecting multifactorial DILI mechanisms alone, as none of the models consider all mechanisms [72]. The scientific community can enable the comparison of many in vitro models, for example, by establishing consensus on reference drugs with recommended test concentrations and exclusion criteria for conducting and interpreting in vitro studies. Any drug testing with a wide variety of options provides insight into the hepatotoxic potential [65,72]. The detection of drug-induced liver toxicity in the assessment of general cytotoxicity can currently only be characterized by a rationalized, multi-stage testing strategy.

Sufentanil and remifentanil reduced vitality and showed negative effects on cell integrity (LDH release in the cell culture supernatant) and/or cell count. According to the literature, the classic side effects of opioids include respiratory depression, nausea, and constipation [73]. However, there is no known hepatotoxic potential [74]. On the contrary, some studies even show a protective effect of sufentanil and remifentanil on the liver in ischemia–reperfusion injury due to anti-inflammatory properties [75,76]. As the hepatocytes used in our assay are tumor cells, it is important to note that there is ongoing research about the anti-tumor effects of sufentanil and inhibition of the proliferation of hepatocellular carcinoma cells (Hep3B) [77]. However, whether sufentanil may show anti-tumor effects remains unclear, and further research is needed.

S-ketamine, the chiral form of ketamine, is a short-acting anesthetic that intensive care physicians widely use due to its favorable hemodynamic, anti-epileptic, opioid-sparing, and bronchodilatory characteristics. Acute side effects include adverse psychological reactions and temporary increases in heart rate and blood pressure [78]. After long-term use, urinary symptoms and hepatotoxicity (hepatic and biliary damage, increased liver enzymes, and even ALF) are known [74,79–81]. In our assay, s-ketamine negatively influenced vitality,

but unlike what we expected, we did not demonstrate a negative effect on the cell count and LDH release. This difference could be because our incubation period was only six days (in contrast to long-term use) or the concentration was not high enough. For concentrations of 100 µg/mL and higher (ten times the highest concentration we tested), Lee et al. showed a reduction in the HepG2 cell count and an increased release of LDH [82].

For propofol, midazolam, partly thiopental, and fentanyl, our results are in line with what we found in the literature: we saw a reduction in hepatocyte vitality without an increase in LDH or a reduction in cell count, which would be consistent with a low probability of hepatotoxicity. None of these drugs is known to be hepatotoxic besides some case reports [83,84]. In addition, higher concentrations of thiopental in plasma led to a marked decrease in the vitality of the test cells. The mechanisms behind this finding are unclear. Thiopental is metabolized in the liver, and the degradation products are eliminated by the kidneys.

Mitochondrial dysfunction plays a key role in the pathophysiology of DILI, and HepG2/C3A cells seem to be an optimal sensor in this regard due to the high content of mitochondria [85]. We found mitochondrial dysfunction (measured by the XTT test) for midazolam and s-ketamin. Midazolam is a short-acting benzodiazepine with sedative and anxiolytic effects. Its spectrum of side effects includes hypotension, agitation, and, especially in combination with opioids, apnea, and hypoxia [86–88]. Our results support the findings of Colleoni et al., who demonstrated a negative effect of midazolam on mitochondrial electron transfer [89] due to its high plasma protein binding of up to 97% [90,91], a normal mitochondrial function (and vitality) was measured in the plasma test as expected. Our results also support the findings of different studies on mitochondrial dysfunction for s-ketamin: it is known to lead to dysfunction of mitochondrial enzymes, e.g., NADH dehydrogenase, and even to mitochondrial degeneration [92–95]. Other than expected, the effect was abolished in plasma, although s-ketamin does have a low plasma protein binding (10–30%) [96]. Although there is some evidence in the literature, we did not find mitochondrial dysfunction for fentanyl and remifentanil. One study by Vilela et al. did show a restriction of mitochondrial function in neuronal cells for both drugs [97]. With HepG2, a slight restriction of the respiratory chain by fentanyl and an improvement by remifentanil were seen [98,99]. For propofol, one major side effect is propofol infusion syndrome. This means the occurrence of heart failure, cardiac arrhythmias, lactic acidosis, renal insufficiency, rhabdomyolysis, hypertriglyceridemia, and hepatomegaly, mainly after long-term use of propofol (>6 days). Even though it is believed to be due to a disturbance of the mitochondrial respiratory chain or fatty acid oxidation [100,101], we did not see mitochondrial dysfunction in HepG2 cells.

HepG2/C3A is capable of synthesizing most plasma proteins, including albumin [41]. For example, albumin production by HepG2/C3A cells is used in the ELAD liver support system [102]. In toxicological studies, albumin synthesis serves as a parameter to assess hepatocyte function: after cell damage, such as after incubation with acetaminophen, microalbumin synthesis is reduced [103]. Our results did not show a significant decrease in the albumin concentration for any of the tested drugs, except for higher concentrations of sufentanil and remifentanil. There is no evidence in the current literature on this topic. Still, a study from 1987 showed a 50% decrease in albumin secretion of human hepatocytes when exposed to different opioids like morphine, heroin, and methadone in higher concentrations [31].

The activity of CYP1A2, one of the most important CYP450 enzymes, especially for catalyzing many reactions involved in drug metabolism, was diminished by s-ketamine and remifentanil. For remifentanil, there is no evidence in the literature regarding the influence on CYP450 enzymes. In the organism, it is exclusively metabolized via tissue-independent esterases, which makes it unique among all opioids with a context-sensitive half-time of 3.5 min, regardless of infusion time [104]. Despite the fact that s-ketamine is metabolized to norketamine via CYP3A4 and CYP2B6, our results show that s-ketamine acts as an inhibitor of the CYP1A2 enzyme. In the literature, there is evidence that ketamine

can act as an inhibitor or inducer of CYP1A2, depending on the route and duration of administration [105]. Two tested drugs, propofol and thiopental, increased the activity of CYP1A2. For propofol, this confirms the results of a study on rabbits [106]. However, studies in human hepatic microsomes showed that propofol appears to be a weak inhibitor of CYP 1A2, 2C9, and 3A4 isoenzymes [107]. Propofol itself is metabolized via CYP2B6 and CYP2C9 [108]. Although the exact degradation pathways for thiopental are unknown, the literature shows that it acts as an inducer of CYP3A3 and 3A4 [109]. An inducing effect on CYP1A2, as we see in our results, is not known. Midazolam, fentanyl, and sufentanil are metabolized almost exclusively by CYP3A4 [110,111]. In agreement with Vrzal et al., we did not detect any effect on CYP1A2 activity for midazolam [112]. For fentanyl and sufentanil, there is no evidence in the literature of the effects of the CYP450 system. All substances are excreted from the urine after metabolism via the before-mentioned CYP enzymes (and glucuronidation for some).

The use of DMSO as a solvent for propofol had no negative effects on cell viability and functionality in the experiments, and no solvent-related effects were observed in the propofol experiments using DMSO as a solvent (data in Supplementary Table S1). DMSO is widely used in biomedical research, toxicology, pharmacology, and cell biology. It is used as a solvent for cryopreservation and can have different effects depending on the concentration and cell type studied. Below a concentration of 10%, DMSO is generally considered non-toxic, but its cytotoxicity is concentration-dependent [113,114]. We could also exclude any pH-related effects on the test results, as the tested pH of all drugs at the $C_{max}$ was in the range of the negative control (data in Supplementary Tables S2 and S3).

Three major limitations in this study could be addressed in further research. First, the metabolic capacity of HepG2/C3A cells is restricted regarding certain enzymes (e.g., cytochrome enzymes), degradation pathways (e.g., plasma esterases), and transport proteins (e.g., bile salt export pump), which could lead to possible over- or underestimation of hepatotoxicity. The used cells are suitable for testing parental substances but not for determining metabolite-dependent toxicity [42]. Second, other cells typical for the liver (e.g., Kupffer cells) are missing in this assay, and some evidence suggests that certain drugs (e.g., diclofenac) lead to hepatoxicity via pro-inflammatory signals by these cells [115]. Third, it should be considered that HepG2/C3A are tumor cells, and an anti-tumor effect could be misinterpreted as possible hepatotoxicity. For propofol and sufentanil, anti-tumor effects via micro-RNAs and cyclooxygenase have been described [116–118].

Despite these limitations, our study demonstrates a moderate hepatotoxic effect of sufentanil, remifentanil, and s-ketamine in a well-characterized model of in vitro hepatotoxicity and biosensing [28,33–40]. Whether these results can be translated into clinical practice should be further observed depending on further studies that may use a more complex cell culture model of the human liver; for instance, by inducing more enzymes (e.g., CYP3A4), [119] and/or the use of co-culture of hepatocytes and Kupffer cells [120] and the use of 3D models and global pharmacovigilance reports. Regarding drug-induced liver injury (DILI), however, little is known about the hepatotoxicity of examined drugs in critically ill patients. The reason for this could be that in the ICU, many factors can result in liver injury: an underlying disease, a complication during treatment (e.g., sepsis), administration of drugs over a long period and sometimes high doses, or the combination of multiple factors.

## 5. Conclusions

Sufentanil, remifentanil, and s-ketamine showed moderate hepatotoxic effects when in vitro. These drugs should be given with caution in patients vulnerable to hepatotoxic drugs, e.g., patients with pre-existing liver disease or liver impairment as part of the underlying disease. Further studies, especially in critically ill patients, are indicated.

**Supplementary Materials:** The following supporting information can be downloaded at: https://www.mdpi.com/article/10.3390/cimb46040189/s1, Table S1: Test results for HepG2/C3A cells after exposure to 0.9% DMSO (dimethyl sulfoxide) compared with pure cell culture medium after 3 days incubation. Values are represented as median and min/max. LDH: lactate dehydrogenase; Table S2: pH for HepG2/C3A cells in the medium after exposure to sedatives and opioids in concentrations $C_{max}$, $C_{5\times}$, and $C_{10\times}$. Values represent the median and 25th/75th percentiles. Significance between negative control and exposure groups is indicated by * $p < 0.05$; Table S3: pH for HepG2/C3A cells in plasma after exposure to sedatives and opioids in concentrations Cmax, C5×, and C10×. Values represent the median and 25th/75th percentiles. The significance between negative control and exposure groups is indicated by * $p < 0.05$.

**Author Contributions:** Conceptualization, K.H., M.S. and S.D.; methodology, S.D. and K.H.; formal analysis, K.H.; investigation, K.H. and S.D.; data curation, K.H., M.S. and S.D., writing—original draft preparation, S.D., K.H. and M.S.; writing—review and editing, M.S. and K.H.; visualization, K.H.; supervision, M.S.; project administration, M.S. All authors have read and agreed to the published version of the manuscript.

**Funding:** This study relied solely on financial resources from the University of Rostock, the European Regional Development Fund (EFRE), and the European Social Fund (ESF), Grant no. AU 09 046:ESF/IV-BM-B35-0005/12.

**Institutional Review Board Statement:** Not applicable.

**Informed Consent Statement:** Not applicable.

**Data Availability Statement:** The data presented in this study are available on request from the corresponding author.

**Acknowledgments:** The author would like to thank Heike Potschka for the excellent work in the past years.

**Conflicts of Interest:** The authors declare no conflicts of interest.

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
