# Peer review of "In Vitro Hepatotoxicity of Routinely Used Opioids and Sedative Drugs"

_cimb, doi:10.3390/cimb46040189_

Round 1

Reviewer 1 Report (New Reviewer)

Comments and Suggestions for Authors

The authors have incubated a permanent hepatic cell line with several analgetics to investigate their potential hepatotoxic. Concentrations used were Cmax in serum obtained during therapeutic i.v. application as well as 5 and 10 times this concentration. At Cmax only one (sufentanil) of the 7 compouns tested showed toxicity at the clinically relevant concentration.

Three drugs (sufentanil, remifentanil and s-ketamin) presented a significant reduction in three or more

of six test parameters, that has been interpreted as a moderate hepatotoxicity.

Several points need to be addressed in a revised manuscript:

·       First of all, what is the reliability of data obtained using an immortalized human hepatoma cell line, which will have different physiological functions than a normal hepatocyte.

·       Table 1 needs to be expanded to understand at what concentration effects have been observed. Moreover, there is no Table which indicates at what concentrations the combined drugs became effective.

·       The Cmax values are determined after i.v. infusion of the drugs. Is this concentration relevant for long term analgetic treatment?

·       EROD activity has been measured after methylcholanthrene induction. What is the relevance of in vitro effects in case of CYP induction? Again, no information is provided at what concentrations the drugs become effective.

The abundant literature (144!) should be reduced.

Without this additional information the relevance of the data presented cannot be evaluated. 

Comments on the Quality of English Language

minor editing

Author Response

Dear Reviewers,

Thank you very much for your review and valuable comments. We hope, we could sufficiently respond to questions. Changes in the manuscript are highlighted in this point-by-point response (in Red and bold) and in the manuscript.

Reviewer 1

The authors have incubated a permanent hepatic cell line with several analgetics to investigate their potential hepatotoxic. Concentrations used were Cmax in serum obtained during therapeutic i.v. application as well as 5 and 10 times this concentration. At Cmax only one (sufentanil) of the 7 compouns tested showed toxicity at the clinically relevant concentration. Three drugs (sufentanil, remifentanil and s-ketamin) presented a significant reduction in three or more of six test parameters, that has been interpreted as a moderate hepatotoxicity. Several points need to be addressed in a revised manuscript

First of all, what is the reliability of data obtained using an immortalized human hepatoma cell line, which will have different physiological functions than a normal hepatocyte.

Answer:
Only sufentanil reduced significantly cell count at the clinically relevant concentration (Cmax) in cell culture medium and plasma. But regarding the results of the other test-parameters, also other tested drugs showed impairment of the function or viability of tested hepatocyte-cells at Cmax- concentrations.

Please see to the last part of the chapter “Introduction (we wrote to the topic” reliability of data”): “We used an established and standardized biosensor [35] with the well-established immortalized human HepG2/C3A cells, a patented, highly functional clonal derivative of the human hepatoma cell line HepG2,to test the hepatotoxicity of routinely used sedatives and opioids. [36–38] The evaluation of multiple endpoints on Hep G2 cells allows the prediction of human hepatotoxicity with a sensitivity of over 80% and a specificity of 90%. [39] Accordingly, HepG2 cells and derivatives can predict general human hepatotoxicity based on hepato-specific endpoints. [23,40] These cell lines are characterized by their functional similarity to human liver, physiological response to toxic insults and metabolic markers. [35,36,41–49]”

The use of immortalized human hepatoma cell lines in research can provide reliable data, but is associated with some limitations. We have described the limitations in the last part on the chapter “Discussion”. In summary, hepatoma cell lines offer advantages in terms of availability, consistency, and their suitability for toxicity studies and modeling specific physiological processes.

Table 1 needs to be expanded to understand at what concentration effects have been observed. Moreover, there is no Table which indicates at what concentrations the combined drugs became effective.

Answer:
Table 1 summarized the results of testing Cmax-concentrations. We agree with the reviewer and have clarified this in the title of table 1:

“Table 1. Comparison of tested sedatives and opioids in medium (M) and in plasma (P) at Cmax (clinically relevant concentration) compared to NC (neg. Ctrl.). The arrows show a significant (P < 0.05 ) increase (↑) or decrease (↓) in the corresponding category compared to the negative control. The number of dots corresponds to the number of arrows of all assays and represents the hepatotoxic potential of the different drugs in medium and plasma.”

The Cmax values are determined after i.v. infusion of the drugs. Is this concentration relevant for long term analgetic treatment?

Answer:
The aim of the study was the testing of hepatotoxicity using clinical relevant concentrations of drugs during continuous analgosedation. We used steady state plasma concentration of the drugs (we have clarified this in the chapters “Introduction” and “Materials and Methods).”

The used Cmax values may not be directly relevant for long-term analgesic treatment.

EROD activity has been measured after methylcholanthrene induction. What is the relevance of in vitro effects in case of CYP induction? Again, no information is provided at what concentrations the drugs become effective.

Answer:
Yes, CYP1A2-activity can be measured in HepG2/C3A cells only after treatment with methylcholanthrene (MCH). MCH is a known inducer of CYP1A2, and the use of HepG2/C3A, a human liver cell line, in combination with MCH enables the induction of CYP1A2. Important is, that for all experiments also the same concentration of MCH was used.

There are few cell-based assays that can determine the induction and inhibition of the phase I enzymes that metabolise drugs, yet these are of central importance.  We report here on an assay for the induction of cytochrome P-450 (CYP) isoenzyme 1A2 that fulfils these requirements. HepG2/C3A and ethoxyresorufin are used. CYP1A2 can be strongly induced with MCH.

Literature:

  1. H. Kelly, N.L. Sussman, A Fluorescent Cell-Based Assay for Cytochrome P-450 Isozyme 1A2 Induction and Inhibition, 2000, Journal of Biomolecular Screening 5(4):249-54
  2. T. Donato, M. J. Gomezlechon, and J. V. Castell, A microassay for measuring cytochrome P450IA1 and cytochrome P450IIB1 activities in intact human and rat hepatocytes cultured on 96-well plates, 1993, Analytical Biochemistry 213(1):29-33

The rich literature (144!) should be reduced.

Answer:
We agree and have reduced the number of references.

We would like to thank all reviewers for their assessment and valuable comments.

Yours sincerely,

Martin Sauer

- for all authors of  the manuscript

Reviewer 2 Report (New Reviewer)

Comments and Suggestions for Authors

Evaluating the hepatotoxicity of opioids and sedative drugs in hepatic cell lines is a very meaningful study and will serve as an important reference for researchers in related fields.

The academic significance of this manuscript will be further strengthened if references to responses in primary cells of liver injury model animals other than HepG2/C3A cells or in other organ-derived cell lines are discussed.

Author Response

Dear Reviewers,

thank you very much for your review and valuable comments. We hope, we could sufficiently respond to questions. Changes in the manuscript are highlighted in this point-by-point response (in Red and bold) and in the manuscript.

Reviewer 2

Evaluating the hepatotoxicity of opioids and sedative drugs in hepatic cell lines is a very meaningful study and will serve as an important reference for researchers in related fields.

The academic significance of this manuscript will be further strengthened if references to responses in primary cells of liver injury model animals other than HepG2/C3A cells or in other organ-derived cell lines are discussed

Answer:
We agree and have discussed the available know results of other cell-models and partly animal-models in the chapter "Discussion".

Our literature-search did not provided any specific information on the use of primary hepatocytes for testing the hepatotoxicity of sedatives and opioids.

Animal models poorly predict drug safety in humans (please see the references 79 and 80 in the paper). We found mainly animals models who described accumulation of opioids in the case of liver failure. For instance:

Kostopanagiotou G, Markantonis SL, Arkadopoulos N, Andreadou I, Charalambidis G, Chondroudaki J, Costopanagiotou C, Smyrniotis V. The effect of acutely induced hepatic failure on remifentanil and fentanyl blood levels in a pig model. Eur J Anaesthesiol. 2006 Jul;23(7):598-604. doi: 10.1017/S0265021506000135

Kostopanagiotou G, Markantonis SL, Arkadopoulos N, et al. The effect of acutely induced hepatic failure on remifentanil and fentanyl blood levels in a pig model. European Journal of Anaesthesiology. 2006;23(7):598-604. doi:10.1017/S0265021506000135

We would like to thank all reviewers for their assessment and valuable comments.

Yours sincerely,

Martin Sauer
- for all authors of  the manuscript

Reviewer 3 Report (New Reviewer)

Comments and Suggestions for Authors

The authors have examined effects of certain CNS depressants/anesthetics on viability, mitochondrial function and metabolic capability in a human hepatocyte cell line.  They recognize the 3 major deficiencies of the experiments--limited metabolic capability, lack of other liver cells, and the cells used are cancer cells.  The impact of the results would be much stronger if any of these had been addressed in the experiments themselves.

There are a lot of words and a lot of references in this article.

The Introduction could be made more concise.  In the Abstract, the authors need to rewrite the first sentences, making sure the reader understands it is the toxicity of sedatives and not the hepatocyte cell line that hasn't been well characterized.

The authors used literature Cmax values to justify the concentrations used for testing.  This is reasonable, although some values are considerably higher than provided in some references not cited (e.g., fentanyl in Goodman & GIlman's Pharmacology).

Figure 1 has been prepared with a font too small to read.  This must be improved so readers know which bar goes with with drug without using a magnifying glass.

Table 1 uses arrows and 'level of hepatotoxicity' which are not as informative an providing just how much (maybe in %) the endpoints changed.  Was the level of hepatotoxicity related to quartiles?  (that is not explained).

The Discussion is very long and repeats some of the Introduction.  The concentration of DMSO used as solvent was low and typical for what is used in cell culture so the paragraph on it could be deleted from this section.

Reference 12 does not talk about alpha-1 agonists in sedative management.  An alpha-2 drug (e.g., clonidine) was mentioned.

Comments on the Quality of English Language

Watch that singular nouns are matched with singular verbs (and plural nouns with plural verbs) at the beginning of the Abstract.

Author Response

Dear Reviewers,

Thank you very much for your review and valuable comments. We hope, we could sufficiently respond to questions. Changes in the manuscript are highlighted in this point-by-point response (in Red and bold) and in the manuscript.

Reviewer 3

The authors have examined effects of certain CNS depressants/anesthetics on viability, mitochondrial function and metabolic capability in a human hepatocyte cell line.  They recognize the 3 major deficiencies of the experiments--limited metabolic capability, lack of other liver cells, and the cells used are cancer cells.  The impact of the results would be much stronger if any of these had been addressed in the experiments themselves.

There are a lot of words and a lot of references in this article.

Answer:
We agree and have the following sentences in the chapters “Introduction” and “Discussion”:

“We used an established and standardized biosensor [35] with the well-established immortalized human HepG2/C3A cells, a patented, highly functional clonal derivative of the human hepatoma cell line HepG2,to test the hepatotoxicity of routinely used sedatives and opioids. [36–38] The evaluation of multiple endpoints on Hep G2 cells allows the prediction of human hepatotoxicity with a sensitivity of over 80% and a specificity of 90%. [39] Accordingly, HepG2 cells and derivatives can predict general human hepatotoxicity based on hepato-specific endpoints. [23,40] These cell lines are characterized by their functional similarity to human liver, physiological response to toxic insults and metabolic markers. [35,36,41–49] “

“HepG2/C3A are capable of synthesizing most plasma proteins, including albumin. [119] Albumin production by HepG2/C3A cells is used, for example, in the ELAD liver support system. [120] In toxicological studies, albumin synthesis serves as a parameter to assess hepatocyte function: after cell damage, such as after incubation with acetaminophen, microalbumin synthesis is reduced. [121]”

“There are three major limitations in this study that could be addressed in further research. First, the metabolic capacity of HepG2/C3A cells is restricted regarding certain enzymes (e.g., cytochrome enzymes), degradation pathways (e.g., plasma esterases) and transport proteins (e.g., bile salt export pump), which could lead to possible over- or underestimation of hepatotoxicity. Second, other cells typical for the liver (e.g., Kupffer cells) are missing in this assay and some evidence suggests that certain drugs (e.g., diclofenac) lead to hepatoxicity via pro-inflammatory signals by these cells. [138] Third, it should be considered that HepG2/C3A are tumor cells and an anti-tumor effect could be misinterpreted as possible hepatotoxicity. For propofol, and sufentanil anti-tumor effects via micro-RNAs and cyclooxygenase have been described. [94,139–142]”

We have reduced the word count in the chapter “Discussion” and the number of references.

The Introduction could be made more concise.  In the Abstract, the authors need to rewrite the first sentences, making sure the reader understands it is the toxicity of sedatives and not the hepatocyte cell line that hasn't been well characterized.

Answer:
We agree and have rewritten the sentence:

"A hepatocyte cell line was used for determination of hepatotoxicity of sedatives and opioids, because the hepatotoxicity of these drugs has not been well characterized yet."

The authors used literature Cmax values to justify the concentrations used for testing.  This is reasonable, although some values are considerably higher than provided in some references not cited (e.g., fentanyl in Goodman & GIlman's Pharmacology).

Answer:
Thanks for the advice. We had selected various sources and found that the C max ranges fluctuate. Cmax values can vary in the literature due to factors such as different formulations of the drug, dosing, route of administration, and individual patient differences. We used the lowest test concentration of the various sedative medications, which are the mean plasma level after initiation of a continuous intravenous therapy (C max, as steady state plasma concentration) based on clinical application data (the chose even the lowest values from the literature) and additionally, the 5x Cmax- and 10x Cmax- concentrations.

For instance – fentanyl – chousen Cmax: 10 pg/ml:

C. S. REILLY, A. J. J. WOOD, M. WOOD. Variability of fentanyl pharmacokinetics in man. Computer predicted plasma concentrations for three intravenous dosage regimens. Anaesthesia 1984; 40:837-843: “The steady state plasma concentration reached with an infusion of 0.3 microgram/kg/minute varied from 12.2-119.9 ng/ml and the plateau level attained with a two-rate infusion (2.7 microgram/kg/minute for 20 minutes then 0.3 microgram/kg/minute) ranged from 10.6-50.8 ng/ml.”

Figure 1 has been prepared with a font too small to read.  This must be improved so readers know which bar goes with with drug without using a magnifying glass.

Answer:
We agree and have re-prepared the figure 1:

If the reviewer satisfied with this solution, we would prepared the figure 1 in this spirit for the print.  

Table 1 uses arrows and 'level of hepatotoxicity' which are not as informative an providing just how much (maybe in %) the endpoints changed.  Was the level of hepatotoxicity related to quartiles?  (that is not explained).

Answer:
In the table, significant changes compared to the negative control were marked with arrows for the corresponding parameters. If there was a significant increase, the arrow pointing upwards was selected; if a significant reduction was determined, an arrow pointing downwards was selected.

The table heading was adapted for clarifying:

“Comparison of the tested sedatives and opioids in the medium (M) and in the plasma (P) at Cmax (clinically relevant concentration) compared to NC (neg. control). The arrows show a significant (P < 0.05)  increase or decrease in the corresponding category compared to the negative control. The number of dots corresponds to the number of arrows of all assays and represents the hepatotoxic potential of the different drugs in medium and plasma.”

The Discussion is very long and repeats some of the Introduction.  The concentration of DMSO used as solvent was low and typical for what is used in cell culture so the paragraph on it could be deleted from this section.

Answer:
We have revised and shortened the chapter „Discussion“ (e.g., the part of DMSO).

Reference 12 does not talk about alpha-1 agonists in sedative management.  An alpha-2 drug (e.g., clonidine) was mentioned.

Answer:
Sorry for this mistake; of course alpha-2-agoist is right. We have revised it in the chapter "Introduction".

We would like to thank all reviewers for their assessment and valuable comments.

Yours sincerely,

Martin Sauer
- for all authors of  the manuscript

Round 2

Reviewer 1 Report (New Reviewer)

Comments and Suggestions for Authors

The authors did not sufficiently respond to the reviewer’s questions and recommendations.

First of all, they did not justify to what extent the hepatoma cells mimic intact hepatocytes. Immortalized cells usually loose certain functions and maintain survival activities. Since they measured mere cytotoxicity any other cell line may have shown the same effects. Since most readers will not be aware of this problem, the authors should briefly describe the capability of the hepatoma cell line in comparison to primary hepatocytes. 

They indicate that albumin production of the cultured hepatocytes was examined. Like LDH albumin release into the medium is an indicator for membrane damage rather than production.

What is the rational and clinical relevance of measuring IROD activity and this after MC induction?

The previous review asked for a justification whether a Cmax concentration after i.p. injection is relevant for long term treatment. The authors responded that this has been addressed in the Introduction and Methods. This is not the case.

Previous recommendation: the rich number of references should be reduced. They reduced it from 144 to 137!!

The general problem of the study is that there is no information on toxicokinetics of the different drugs, their metabolism and excretion so that a mere determination of cytotoxicity is of little relevance and even misleading. At least this needs to be addressed in the discussion.

Comments on the Quality of English Language

Minor editing needed

Author Response

Dear Reviewer,

thank you very much for your review and valuable comments. We hope, we could sufficiently respond to questions. Changes in the manuscript are highlighted in this point-by-point response (in Red and bold) and in the manuscript.

The authors did not sufficiently respond to the reviewer’s questions and recommendations.

First of all, they did not justify to what extent the hepatoma cells mimic intact hepatocytes. Immortalized cells usually loose certain functions and maintain survival activities. Since they measured mere cytotoxicity any other cell line may have shown the same effects. Since most readers will not be aware of this problem, the authors should briefly describe the capability of the hepatoma cell line in comparison to primary hepatocytes.

Answer:
We have added the following sentences in the chapter "Introduction":

HepG2/C3A are able to synthesize most plasma proteins, including albumin. [41] However, their main disadvantage is their lower metabolic capacity due to low lev-els of CYP enzymes compared to primary isolated hepatocytes. This makes them suitable test models for the parent substances, but not for the toxicity of metabolites. [42] On the other hand, primary isolated human hepatocytes are poorly available, have a different consistency and CYP-induction varying from donor to donor and may dependent from medication of the patient.

Our literature-search did not provided any specific information on the use of primary hepatocytes for testing the hepatotoxicity of sedatives and opioids.

They indicate that albumin production of the cultured hepatocytes was examined. Like LDH albumin release into the medium is an indicator for membrane damage rather than production.

Answer:
We disagree; we have used the secretion of albumin as an established test-parameter, described in some studies. Please see literature (enclosed in the paper):

  1. Bouma, M.E.; Rogier, E.; Verthier, N.; Labarre, C.; Feldmann, G. Further cellular investigation of the human hepatoblastoma-derived cell line HepG2: morphology and immunocytochemical studies of hepatic-secreted proteins. In Vitro Cell. Dev. Biol. 1989, 25, 267–275, doi:10.1007/BF02628465.
  2. Millis, J.M.; Cronin, D.C.; Johnson, R.; Conjeevaram, H.; Conlin, C.; Trevino, S.; Maguire, P. Initial experience with the modified extracorporeal liver-assist device for patients with fulminant hepatic failure: system modifications and clinical impact. Transplantation 2002, 74, 1735–1746, doi:10.1097/00007890-200212270-00016.
  3. Ullrich, A.; Berg, C.; Hengstler, J.G.; Runge, D. Use of a standardised and validated long-term human hepatocyte culture system for repetitive analyses of drugs: repeated administrations of acetaminophen reduces albumin and urea secretion. Altex 2007, 24, 35–40, doi:10.14573/altex.2007.1.35.

What is the rational and clinical relevance of measuring IROD activity and this after MC induction?

Answer:
CYP1A2-activity can be measured in HepG2/C3A cells only after treatment with methylcholanthrene (MCH), because the activity in „pure“ cells is to low. Important is, that for all experiments in our study also the same concentration of MCH was used. The relevance of measuring  the activity of CYP1A2 is that the activity is an established parameter in toxicological studies and is clinically very important for the metabolism of many drugs. CYP1A2 is one of the most important cytochrome P450 (CYP) enzymes in the liver, accounting for 13% to 15% of hepatic CYP enzymes.

The previous review asked for a justification whether a Cmax concentration after i.p. injection is relevant for long term treatment. The authors responded that this has been addressed in the Introduction and Methods. This is not the case.

Answer:
We are sorry, but the reviewer asked for concentration relevant for long term analgetic treatment. We answered, that the used Cmax values may not be directly relevant for long-term analgesic treatment.

Intraperitoneal (IP) route of drug administration in laboratory animals is a common practice in many in vivo studies of disease models, but this method is not relevant for clinical practise in analagosedation in humans (and for toxicologocal studies in this field).

Previous recommendation: the rich number of references should be reduced. They reduced it from 144 to 137!!

Answer:
We reduced it now  to 121. In our opinion a further reduction of references are not possible, because we discuss many drugs and methods.

The general problem of the study is that there is no information on toxicokinetics of the different drugs, their metabolism and excretion so that a mere determination of cytotoxicity is of little relevance and even misleading. At least this needs to be addressed in the discussion.

Answer:
We had discussed some aspects of metabolism in the chapter "Discussion" (yellow-marked). For improving this part of discussion we added now the following sentences (bold) in two different parts:

As drugs for sedation and analgesia are given over a long period of time, the question of possible hepatotoxicity arises here in particular. Additionally, all tested sedatives and opioids are hepatically metabolized mostly via CYP with the exception of remifentanil, which can lead to accumulation of these drugs in the course of liver in-sufficiency. On the other side, ICU patients need often 8 to 12 different drugs and possible toxic interaction and summarization of used compounds may lead to a higher risk of organ toxicities.

Activity of CYP1A2, one of the most important CYP450 enzymes especially for catalyzing many reactions involved in drug metabolism, was diminished by s-ketamine and remifentanil. For remifentanil there is no evidence in literature regarding the influence on CYP450 enzymes. In the organism, it is exclusively metabolized via tissue-independent esterases, which makes it unique among all opioids with a context-sensitive half-time of 3.5 minutes, regardless of infusion time. [105] Despite that fact that s-ketamine is metabolized to norketamine via CYP3A4 and CYP2B6 our results show that s-ketamine acts as an inhibitor of the CYP1A2 enzyme. In literature there is evidence that ketamine can act as an inhibitor or inducer of CYP1A2, depending on route and duration of administration. [106] Two tested drugs, propofol and thiopental, increased the activity of CYP1A2. For propofol this confirms the results of a study in rabbits. [107] However, studies in human hepatic microsomes showed that propofol appears to be a weak inhibitor of CYP 1A2, 2C9, and 3A4 isoenzymes. [108] Propofol itself is metabolized via CYP2B6 and CYP2C9. [109] Although for thiopental the exact degradation pathways are not known, literature gives evidence that it acts as an inducer of CYP3A3 and 3A4. [110] An inducing effect on CYP1A2 like we see in our results is not known. Midazolam, fentanyl and sufentanil are metabolized almost exclusively by CYP3A4. [111,112] In agreement with Vrzal et al., we did not detect any effect on CYP1A2 activity for midazolam. [113] For fentanyl and sufentanil there is no evidence in literature of effects the CYP450 system. After metabolism via the before mentioned CYP enzymes (and glucuronidation for some) all substances are excreted with the urine.

Round 3

Reviewer 1 Report (New Reviewer)

Comments and Suggestions for Authors

After all, the authors do not consider the limited metabolic capacity of the cell line used. Although they state that the cells only express CYP1A2 they describe that most of the drugs used are metabolized by other CYPs, which are not or only weakly expressed in the cells and even describe that some undergo glucuronidation, which if expressed glucuronidation is of minor activity due to lack of cofactors. All this questions the relevance of the findings. To avoid the impression that the authors just incubated a cell line with certain drugs and describe the outcome without considering the enzymatic capability of the cells and the specific metabolism of the compounds, the authors should clearly describe the limitations and with this the relevance of their findings for the intact organism in the discussion and briefly in the abstract.

Comments on the Quality of English Language

Minor editings.

Author Response

Dear Reviewer,

Thank you very much for your review and valuable comments. We hope, we could sufficiently respond to questions. Changes in the manuscript are highlighted in this point-by-point response (in Red and bold) and in the manuscript.

Reviewer 1

After all, the authors do not consider the limited metabolic capacity of the cell line used. Although they state that the cells only express CYP1A2 they describe that most of the drugs used are metabolized by other CYPs, which are not or only weakly expressed in the cells and even describe that some undergo glucuronidation, which if expressed glucuronidation is of minor activity due to lack of cofactors. All this questions the relevance of the findings. To avoid the impression that the authors just incubated a cell line with certain drugs and describe the outcome without considering the enzymatic capability of the cells and the specific metabolism of the compounds, the authors should clearly describe the limitations and with this the relevance of their findings for the intact organism in the discussion and briefly in the abstract.

Answer:

We hope, that we could now more clarify the limitation of our cell-model in the revised manuscript:

We added following sentences in the abstract, in the chapter “Introduction” and “Discussion”:

Abstract:

Further studies are indicated to this topic may using more complex cell culture model and global pharmacovigilance report, addressing the limitation of the used cell-model:  HepG2/C3A have a lower metabolic capacity due to low levels of CYP enzymes compared to primary hepatocytes; however, the test-model is suitable for parental substances, but not for the toxicity-testing of metabolites.

Introduction:

HepG2/C3A are able to synthesize most plasma proteins, including albumin. [41] However, their main disadvantage is their lower metabolic capacity due to low levels of CYP enzymes compared to primary isolated hepatocytes. This makes them suitable test models for the parental substances, but not for the toxicity of metabolites. [42] HepG2/C3A are able to synthesize most plasma proteins, including albumin. [41] They produce bile acids as well as glycogen and express many hepatic functions, such as cholesterol and triglyceride metabolism, lipoprotein metabolism or insulin signaling. [42–44] Differences are, however a non-functional urea cycle, low levels of phase-II enzymes (sulfotransferase, uridine diphosphate glucuronosyltransfer-ase, glutathione S-transferase, or N-acetyltransferase) and transport proteins (o-ganic anion trans-porting polypeptide C, bile salt export pump, and sodi-um-taurocholate co-transporting polypeptide). [45] Although there are low or absent basal levels of important CYP enzymes (such as CYP3A4, CYP2C9, CYP2C19, CYP2A6, or CYP2D6) compared to primary hepatocytes [46], a similar inducibility has been shown for CYP 1A1, 1A2, 2B6, and 3A4. [44, 47]

Discussion:

There are three major limitations in this study that could be addressed in further research. First, the metabolic capacity of HepG2/C3A cells is restricted regarding certain enzymes (e.g., cytochrome enzymes), degradation pathways (e.g., plasma esterases) and transport proteins (e.g., bile salt export pump), which could lead to possible over- or underestimation of hepatotoxicity. The used cells are suitable for testing parental substances, but not for the determination of metabolites dependent  toxicity. [42]

In our model we have measured the direct toxicity of parental drugs. We agree with the reviewer, that HepG2/C3A cells not represent all biotransformation capacity like origin hepatocytes. On the other hand, C3A cells are widely used for toxicological testing to estimate the hepatotoxic potential of drugs and chemicals and for an extracorporeal liver assist device. Additionally, we have validated and used our “broad-spectrum” hepatotoxicity-test in some clinical and experimental studies.

This manuscript is a resubmission of an earlier submission. The following is a list of the peer review reports and author responses from that submission.

Round 1

Reviewer 1 Report

Comments and Suggestions for Authors

The study presents valuable insights into the hepatotoxicity of commonly used sedatives and opioids in the ICU. However, the manuscript would benefit from a clearer articulation of its objectives, more detailed methodological descriptions, and a more thorough analysis and discussion of its findings. Additionally, considering the potential clinical implications and limitations would significantly enhance the impact and relevance of the study.

1. Abstract:

Query: The abstract succinctly summarizes the study but lacks a clear statement of its objective. Could you explicitly state the primary research question or hypothesis at the beginning of the abstract?

Suggestion: Add a brief mention of the methodology (i.e., the use of HepG2/C3A cells) early in the abstract for context.

Improvement Needed: The implications of the findings for clinical practice are somewhat vague. Could this be clarified or expanded upon?

2. Introduction:

Query: The introduction effectively sets the stage for the research, but could benefit from a more detailed explanation of why understanding the hepatotoxicity of these drugs is particularly critical in the ICU setting.

Suggestion: Include a brief overview of existing literature on the topic, highlighting the gap your study aims to fill.

3. Materials and Methods:

Improvement Needed: The description of cell culture methods and drug preparation is clear, but could be improved with more detail on the control setups used for comparison.

Query: Is there a specific reason for selecting the concentrations of drugs tested? A rationale for these choices would strengthen this section.

4. Results:

Suggestion: Consider presenting the results in a more organized manner, possibly with subheadings for clarity. This could help readers quickly grasp the key outcomes for each drug tested.

Improvement Needed: The presentation of statistical analysis appears to be missing. Providing details on statistical methods and significance levels would be beneficial.

5. Discussion:

Improvement Needed: The discussion could be enhanced by directly relating the findings to potential clinical implications and comparing them with existing studies.

Suggestion: Address potential limitations of your study and how they might impact the interpretation of the results.

Reviewer 2 Report

Comments and Suggestions for Authors

In this study, Haller et al. investigate the hepatotoxicity of various drugs. This study lacks novelty and appropriate experimental design.

·       The authors claim to study hepatotoxicity of drugs, but they use HepG2 cells, which is well-known hepatocellular carcinoma cells. Effects of drugs on HepG2 cells do not show hepatotoxicity. If the authors want to study hepatotoxicity, they should use animal models and determine the effects of interest, such as midazolam, by detecting liver damage, such as ALT, AST, or ALP levels or histological staining. However, there are many previous studies for that. For example, it is known that midazolam does not elevate ALT levels in the short duration or low dose. This study lacks novelty for hepatotoxicity of those drugs.

·       Doses of drugs are determined already in clinical situations, and it is not reasonable to study far higher dose in this study. For example, dose of midazolam is generally 0.1-0.25 mg/kg, so 6-15 mg for a 60 kg person. We know that these doses are effective and do not cause hepatotoxicity. In this study, the authors use 100 mg as Cmax and 5xCmas and 10xCmax for midazolam. It does not make sense why the authors study these high concentrations. These procedures do not mimic clinical procedures for human patients.

·       Data shown by authors here, such as Figure 1, are anti-cancer effects of those drugs, because the authors use HepG2 cells. Do the authors want to seek anti-cancer effects of those drugs? There are many previous studies for that. For example, multiple previous studies demonstrated that midazolam shows anti-cancer effects on various cancers including hepatocellular carcinoma. This study still lacks novelty if the authors want to show anti-cancer effects. Overall, this study lacks the rationale and novelty with insufficient literature search for experimental design. The issues and flaw of this study cannot be fixed by revision, so this study cannot be accepted for publication.